# Prolonged Piezo1 Activation Induces Cardiac Arrhythmia

**DOI:** 10.3390/ijms24076720

**Published:** 2023-04-04

**Authors:** Laura Rolland, Angelo Giovanni Torrente, Emmanuel Bourinet, Dounia Maskini, Aurélien Drouard, Philippe Chevalier, Chris Jopling, Adèle Faucherre

**Affiliations:** 1Institute of Functional Genomics, University of Montpellier, CNRS, INSERM, LabEx ICST, 34094 Montpellier, France; 2Neuromyogene Institut, Claude Bernard University, Lyon 1, 69008 Villeurbanne, France; 3Service de Rythmologie, Hospices Civils de Lyon, 69500 Lyon, France

**Keywords:** cardiac arrhythmia, mechanoelectric feedback, *PIEZO1* channel

## Abstract

The rhythmical nature of the cardiovascular system constantly generates dynamic mechanical forces. At the centre of this system is the heart, which must detect these changes and adjust its performance accordingly. Mechanoelectric feedback provides a rapid mechanism for detecting even subtle changes in the mechanical environment and transducing these signals into electrical responses, which can adjust a variety of cardiac parameters such as heart rate and contractility. However, pathological conditions can disrupt this intricate mechanosensory system and manifest as potentially life-threatening cardiac arrhythmias. Mechanosensitive ion channels are thought to be the main proponents of mechanoelectric feedback as they provide a rapid response to mechanical stimulation and can directly affect cardiac electrical activity. Here, we demonstrate that the mechanosensitive ion channel *PIEZO1* is expressed in zebrafish cardiomyocytes. Furthermore, chemically prolonging *PIEZO1* activation in zebrafish results in cardiac arrhythmias. indicating that this ion channel plays an important role in mechanoelectric feedback. This also raises the possibility that *PIEZO1* gain of function mutations could be linked to heritable cardiac arrhythmias in humans.

## 1. Introduction

Cardiac arrhythmias encompass a wide range of abnormalities with a prevalence of 1.5% to 5% in the general population. Among these arrhythmic disorders, atrial fibrillation is the most common, and is frequently associated with high morbidity and mortality [1]. However, the origins and clinical apparition of cardiac arrhythmias can be diverse, as evidenced by the variety of different genes that have been linked to these conditions.

Mechanical forces play a critical role in a large variety of biological processes. Various cellular sensors exist and are present in most organs, including the heart, where they have been shown to be involved in mechanoelectrical feedback, a mechanism that regulates cardiac electrical activity in response to mechanical force [2]. Among the mechanosensors present in the heart, ion channels play a central role in this process as they can both sense mechanical forces and trigger electrical responses. Several reports demonstrate the expression of different mechanosensitive ion channels within the heart, in particular in sinoatrial node (SAN) cells, which are responsible for generating cardiac rhythm [3]. This is also the case for cardiomyocytes in general, where mechanosensitive ion channels can regulate AP duration [2].

Although the underlying mechanisms remain unknown, the presence of mechanosensitive ion channels in cardiomyocytes and the potential role of mechanical forces in regulating cardiac rhythm have frequently been associated with the generation of cardiac arrhythmias. For example, blocking mechanosensitive ion channels with GsMTx4, a peptide isolated from spider venom, can alleviate atrial fibrillation in rabbit models [4]. This also suggests that overactivation of mechanosensitive ion channels could play a causal role in cardiac arrhythmias.

PIEZO1 is a mechanosensitive ion channel present in the plasma membrane that can respond to increases in mechanical stretch, resulting in an intracellular influx of cations, which can ultimately regulate a variety of biological responses [5]. Recently, it has been demonstrated that PIEZO1 is involved in Ca^2+^ homeostasis and signalling in mouse cardiomyocytes, and that loss of *PIEZO1* specifically in cardiomyocytes leads to impaired cardiac function [6]. PIEZO1 can be detected in the sarcolemma, the cardiomyocyte equivalent of the cell plasma membrane. Treating isolated cardiomyocytes with Yoda1, a chemical activator of PIEZO1, results in an increase in intracellular Ca^2+^, indicating that PIEZO1 can modulate cardiomyocyte Ca^2+^ levels. Ca^2+^ is a crucial regulator of cardiomyocyte physiology, and as such, modulating the intracellular concentration of this molecule can produce a spectrum of responses, both physiological and pathophysiological. Cardiomyocyte-specific knockout (KO) of *Piezo1* appears to impact cardiac contractility, resulting in a decreased ejection fraction and fractional shortening in mice. Conversely, cardiomyocyte-specific overexpression of *Piezo1* induces severe heart failure and ventricular tachycardia, indicating that PIEZO1 may also be associated with cardiac arrhythmias [6]. However, in human patients suffering from cardiac arrhythmias, it is unlikely that this condition is caused by an overexpression of *PIEZO1*, specifically in cardiomyocytes. A more logical scenario is that certain genetic variants in *PIEZO1* could lead to a gain of function and subsequent cardiac arrhythmia. Indeed, gain-of-function variants in *PIEZO1* have been identified in patients who suffer from xerocytosis, a rare hemolytic anemia that affects red blood cell physiology [7]. Although no data are available at present regarding the prevalence of cardiac arrhythmias in patients suffering from this condition, mice have been engineered to contain a specific gain-of-function *PIEZO1* mutation found in these patients. In particular, these animals develop cardiomyocyte hypertrophy and widespread cardiac fibrosis. However, in contrast to the tachycardia observed following the overexpression of *Piezo1* in cardiomyocytes, no observed difference in cardiac arrythmias was detected in mice harbouring gain-of-function *Piezo1* mutations [8], although it should be noted that mice in general are unreliable models of cardiac arrythmias, and in particular atrial fibrillation [9,10]. To further elucidate the role of PIEZO1 in cardiac physio/pathophysiology, we opted to utilize a global Piezo1 activation strategy using zebrafish larvae as an in vivo animal model. Zebrafish have proven to be a useful model to study cardiac development and disease [11]. Indeed, compared with mice, zebrafish cardiac electrophysiology and heart rate are more similar to humans [12]. Here, we demonstrate, using a combination of pharmacology and an array of different cardiac analyses, that prolonged Piezo1 activation results in significant cardiac arrhythmias in zebrafish.

## 2. Results

### 2.1. piezo1 is Expressed in Zebrafish Cardiomyocytes

Previous studies in mammals indicate that *Piezo1* is expressed in cardiomyocytes and plays a critical role in Ca^2+^ homeostasis and signaling [6]. To determine whether *piezo1* is expressed in zebrafish cardiomyocytes, we performed a transcriptomic analysis at the single-cell level on adult zebrafish hearts. Because cardiomyocytes are largely absent from single-cell data (their size and fragility are not compatible with scRNAseq pipelines) [13] and since ion channels in general are not highly expressed and are often difficult to detect using standard techniques such as in situ hybridization or immunohistochemistry, we adopted a single-nuclei RNA sequencing (snRNAseq) strategy. Unbiased clustering of ventricular nuclei revealed five distinct clusters (Figure 1A). Each cluster corresponded to a distinct cell type and included *tnnt2*^+^ cardiomyocytes, *spock3*^+^ endothelial cells, *itga2b*^+^ thrombocytes, *csf1ra*^+^ myeloid cells, and *tbx18*^+^ fibroblasts (Figure 1B). Next, we analysed this data set for the expression of the two zebrafish *piezo1* othologs-*piezo1a* [14] and *piezo1b* [15]. In this manner, we were able to detect expression of both *piezo1* orthologs within the cardiomyocyte population (Figure 1C). This data indicates that like in mammals, *piezo1* is expressed in zebrafish cardiomyocytes.

### 2.2. Yoda1 Increases Zebrafish Piezo1 Activation Kinetics

The chemical compound Yoda1 was previously identified as an activator of mammalian PIEZO1 ion channels, affecting the mechanosensitivity and inactivation kinetics [16]. To determine whether Yoda1 also had a similar effect on the zebrafish *PIEZO1* orthologs, we performed electrophysiology on HEK293T cells transfected with either zebrafish *piezo1a* or *piezo1b*. Using a cell-attached configuration in conjunction with negative pressure pulses, we could readily detect Piezo1a inward currents in response to mechanical stimulation (Figure 2A). The current–pressure relationship obtained in the cell-attached configuration revealed an average P_50_ of −13.967 mmHg, suggesting zebrafish Piezo1a has a higher sensitivity to mechanical stimulation compared to previous reports for mammalian PIEZO1 (Figure 2B,C). Next, *piezo1a*-transfected cells were treated with 30 µM of Yoda1, a concentration that has previously been shown to activate mammalian PIEZO1 channels following mechanical stimulation [17]. In this manner, we found that Yoda1 had a similar effect on zebrafish Piezo1a, resulting in a delayed inactivation of the currents in comparison to nontreated cells, which did not significantly affect the total current, as illustrated by the area under the curve (Figure 2B–E). Interestingly, and unlike its mammalian orthologs, Piezo1a sensitivity to mechanical stimulation was not affected by Yoda1 treatment. While Piezo1a mechanically activated currents were easily detected in cell attached configuration, Piezo1b currents were detected in a minority of cells under the same conditions despite being expressed at similar levels to *piezo1a* (Figure 2F and Appendix A). To overcome this shortcoming, we switched to a whole-cell approach in conjunction with a cell-poking piezo device, which revealed a small, mechanically activated inward current in cells expressing *piezo1b* compared to *piezo1a* (Figure 2G) [16,18]. Next, we sought to confirm the effect of Yoda1 treatment on Piezo1a using the whole-cell configuration. Interestingly, in addition to a delayed inactivation, as observed in the cell-attached recordings, we also noticed a deactivation current following the end of the mechanical stimulation in Yoda1-treated cells that was not detected in untreated cells (Figure 2G–I). Altogether, these results indicate that both zebrafish Piezo1 orthologs are mechanically activated and that Yoda1 enhances Piezo1a activation kinetics.

### 2.3. Prolonged Piezo1 Activation In Vivo Causes Cardiac Arrhythmia

We next investigated whether prolonged Piezo1 activation could play a role in cardiac physiology. To achieve this, we treated 4-days-postfertilization (dpf) transgenic *myl7*:EGFP zebrafish larvae with 50 µM Yoda1. The transgenic *myl7*:EGFP zebrafish line expresses GFP specifically in cardiomyocytes, which allows us to record high-definition movies and perform accurate downstream analyses. Following Yoda1 treatment, larvae were mounted in low-melting-point agarose and 1 min movies of the beating heart were captured with a high-speed camera. Movies were subsequently analysed using ZeCardio^TM^ software (Appendix A). In this manner, we were able to determine that treatment with Yoda1 resulted in significant bradycardia, which affected both the ventricle and atrium (Figure 3A,B). We also observed a significant increase in the estimated QT interval (Figure 3D) and the prevalence of atrial fibrillation (no significant arrhythmia was observed in the ventricle) (Figure 3F). Lastly, the end diastolic ventricular diameter appears to be significantly larger than controls following Yoda1 treatment (Figure 3G).

To determine whether this is conserved during adulthood, we performed electrocardiogram (ECG) recordings on adult zebrafish in the exposed heart configuration [19]. After a baseline recording, fish were treated with either DMSO or 50 µM Yoda1 for one hour. A second ECG recording was then performed and compared to the nontreated recording (Figure 4A). We observed that while DMSO had no effect, Yoda1 treatment significantly slowed the heart rate (Figure 4B,C). This bradycardia was associated with an increased PR duration, suggesting a conduction defect between the atrium and ventricle, such as a first-degree atrioventricular block (Figure 4D,E). Taken together, these data indicate that Yoda1 treatment affects heart performance and indicates that prolonged Piezo1 activation results in cardiac arrhythmia.

### 2.4. Prolonged Piezo1 Activation In Vivo Affects the Dynamics of the Cardiac Action Potential

To confirm that prolonged Piezo1 activation results in cardiac arrhythmia, we adopted an optical mapping approach combined with a voltage-sensitive dye to assess the spatiotemporal dynamics of excitable events in the whole heart in vivo [20,21] (Figure 5A and Appendix A). After prolonged incubation with 50 µM Yoda1 or DMSO followed by an incubation in the voltage-sensitive dye, our recordings indicated that Yoda1 significantly affected the heart rate, as evidenced by a reduced number of cardiac APs measured in the ventricular region (Figure 5B). We also observed that heart rhythmicity was affected, as Yoda1 treatment induced an increase in the coefficient of variability of the AP intervals (Figure 5C). Moreover, we noticed that the reduction in heart rate (cardiac APs) and the increase in coefficient of variability of the AP intervals under Yoda1 treatment were caused by a remarkable prolongation in the depolarization phases that occurred in the ventricular region, but not in the atria (Figure 5A and Appendix A). Altogether, these results suggest that Piezo1 activation by Yoda1 impairs heart rhythmicity.

## 3. Discussion

Despite decades-old evidence for the association of cardiac rhythm and mechanosensitivity, the molecular mechanisms of this phenomenon have remained elusive [2]. In particular, mechanosensitive ion channels, such as Piezo1, have long been proposed as a source of mechanoelectric feedback under physiological and pathophysiological conditions. The very nature of mechanosensitive ion channels ensures that they can respond rapidly to changes in mechanical force and translate these into electrical signals that can modulate cardiac performance. Pathological conditions that disrupt the normal cardiac mechanophysiology, for example, myocardial infarction, could affect mechanoelectric feedback and potentially result in cardiac arrhythmias. Recent evidence indicates that PIEZO1 plays a significant role in cardiomyocyte Ca^2+^ handling in response to mechanical stretch, and could be an important factor in the generation of cardiac arrhythmias [6]. To further address the role PIEZO1 plays in cardiac physio/pathophysiology, we chose to expand the current list of cardiac PIEZO1 animal models to include zebrafish. Our data indicate that both *PIEZO1* orthologs (*Piezo1a* and *Piezo1b*) are expressed in zebrafish cardiomyocytes, similar to reports in mammals, where they might play a role in regulating mechanoelectric feedback [22]. Electrophysiological analysis indicates that Piezo1a responds to mechanical stimulation and is activated by the selective agonist Yoda1, resulting in the prolonged activation of the channels in vitro. Using this same approach, we were only able to record a weak Piezo1b current. This could reflect a difference in ion selectivity or suggest that the use of HEK293T-transfected cells does not recapitulate the membrane environment essential for Piezo1b activity [23]. Taken together, these data indicate that the zebrafish Piezo1 orthologs are bona fide mechanosensitive channels that are expressed in cardiomyocytes, and could therefore play a role in mechanoelectric feedback, similar to reports in mammals. Lastly, we demonstrate that prolonging Piezo1 activity in zebrafish larvae affects cardiac rhythm, resulting in a significant decrease in heart rate in conjunction with atrial arrhythmias. Based on the pioneering work of Frank, Starling, and Anrep, it is apparent that the heart is highly responsive to changes in mechanical load [24,25]. Increased myocardial stretch caused by ventricular volume overload results in an elevated force of contraction coupled with cardiomyocyte hypertrophy in order to maintain circulatory homeostasis. The Frank–Starling mechanism relies mainly on the increased Ca^2+^ sensitivity of sarcomeric filaments, while the slow force response (SFR) described by Anrep is contingent with Ca^2+^ transients generated by persistent volume overload [26]. While this response is primarily adaptive, under pathological conditions of sustained volume overload the response becomes maladaptive, resulting in pathological hypertrophy and heart failure. A variety of mechanosensors have been associated with these mechanisms, including mechanosensitive ion channels, which can provide a rapid response to volume overload. For example, the transient receptor potential (TRP) channels TRPC3 and TRPC6 are involved in volume-overload-induced pathological hypertrophy, and are also necessary for generating the Ca^2+^ transients observed in the SFR [27]. Recently, it has been demonstrated that PIEZO1 is also a bona fide cardiomyocyte mechanosensor in vivo, where it translates the mechanical forces exerted on cardiomyocytes into intracellular Ca^2+^ signalling which, ultimately regulates cardiac performance [6]. It is also apparent that increased PIEZO1 signalling results in detrimental effects on cardiac rhythm. In particular, overexpression of *Piezo1* in mouse cardiomyocytes results in cardiac tachycardia. PIEZO1 also instigates the pathological cardiomyocyte hypertrophic response to pressure overload. The initial mechanically induced PIEZO1 response modulates the activity of TRPM4, a stretch-insensitive Ca^2+^ activated ion channel. TRPM4 subsequently activates the Ca^2+^/CALMODULIN-DEPENDENT KINASE II (CaMKII) signalling pathway, which triggers the hypertrophic response in cardiomyocytes. Intriguingly, it appears that because of a close interaction between PIEZO1 and TRPM4, the initial Ca^2+^ influx generated by PIEZO1 subsequently activates TRPM4, which in turn triggers the CaMKII cascade. It appears then that TRPM4 can function as a molecular amplifier for PIEZO1 signalling. Future studies will be required to determine whether a similar mechanism is also associated with the Yoda1-induced arrhythmias we observed. For example, it would be interesting to see whether Yoda1 also affects cardiac rhythm in *trpm4* KO zebrafish larvae. Although much focus has been placed on mechanoelectric feedback in the ventricle, similar mechanisms are also at play in the atria. Atrial fibrillations have long been associated with increased atrial stretch and dilatation, and as such, mechanosensitive ion channels such as PIEZO1 could be involved in this pathology [28]. Atrial fibrillations are caused by afterdepolarization events, which disrupt the normal cardiac action potential. Early afterdepolarizations (EADs) occur during phases 2 (plateau) and 3 (rapid repolarization) of the cardiac action potential prior to the complete repolarization, resulting in an extra cardiac contraction. Delayed afterdepolarizations (DADs) occur after repolarization and prior to the generation of another action potential; this also results in an extra cardiac contraction [29]. Acute atrial stretch is capable of inducing both EADs and DADs. DADs in particular are frequently associated with increased intracellular Ca^2+^, a phenomenon that could potentially be regulated by PIEZO1 [28]. Evidence for this hypothesis is provided by the observation that GsMtx-4, a peptide isolated from tarantula venom, can suppress stretch-induced atrial fibrillations [4]. Although GsMtx-4 targets a number of different mechanosensitive ion channels, it is a potent PIEZO1 inhibitor, which suggests that this ion channel could be involved in the generation of atrial fibrillations in response to stretch. Extrapolating this to our own findings, it is not difficult to imagine that Yoda1-induced extended PIEZO1 activation could mimic the molecular events associated with increased stretch, which may explain the atrial arrhythmias we observed in zebrafish larvae treated with Yoda1.

Lastly, we also observed a significant bradycardia in zebrafish larvae treated with Yoda1. Although the QT interval also increased, this measurement was an estimation based on ventricular contractions, and as such we cannot determine whether or not this is independent of the observed bradycardia. Nevertheless, the decrease in heart rate is very different to the tachycardia observed in mice overexpressing Piezo1 in their cardiomyocytes [6]. However, in this context, the bradycardia we observed may also be caused by activating PIEZO1 in noncardiomyocytes, which may explain the discrepancies between our global Piezo1 activation model and the specific overexpression of *Piezo1* in cardiomyocytes. Bradycardia can be caused by a variety of different conditions, such as aging, heart failure, and myocardial infarction. Obviously, we do not expect any of these conditions to have occurred in Yoda1-treated zebrafish larvae. However, one distinct possibility could be reflex bradycardia. The baroreceptor reflex is responsible for maintaining blood pressure homeostasis. In response to increased arterial tension, the baroreceptor reflex induces bradycardia to counteract the elevated blood pressure. Baroreceptors are mechanosensitive neurons that can detect changes in mechanical stretch exerted on vessel walls by blood pressure and reduce the heart rate accordingly [30]. Recently, it has been demonstrated that both PIEZO1 and PIEZO2 are important components of the baroreceptor mechanosensing required for triggering reflex bradycardia in response to increased arterial blood pressure [30]. In this respect, it is possible that activating PIEZO1 with Yoda1 could trigger the baroreflex and ultimately result in the observed bradycardia.

## 4. Materials and Methods

### 4.1. Zebrafish Husbandry

Zebrafish were maintained under standardized conditions [31], and experiments were conducted in accordance with local approval (APAFIS#2021021117336492-32684) and the European Communities directive 2010/63/EU. The Tg(*myl7*:EGFP) was provided by the CMR[B] Centro de Medicina Regenerativa de Barcelona.

### 4.2. Cell Culture and Transfection

HEK293T cells were maintained in Dulbecco’s Modified Eagle Medium (DMEM) containing 4.5 g/L D-Glucose, Pyruvate, 10% fetal bovine serum, 1% Penicillin/Streptomycin (stock solution—10,000 Units/mL Penicillin, 10,000 mg/mL Streptomycin). For patch clamp experiments, cells were plated onto 35 mm dishes and transfected using Polyplus JetPEI (Ozyme) 24 h after passaging, with 1.5 µg of plasmid for each dish. Cells were analysed 48–72 h after transfection. The plasmids used in this study, pIRES2-GFP-zf*PIEZO1*a and pIRES2-GFP-zf*PIEZO1*b were generated by Genscript.

### 4.3. Patch Clamp Recordings

For cell-attached recordings, HEK293T cells were kept in a bath solution containing (in mM) 140 KCl, 10 HEPES, 1 MgCl_2_, and 10 Glucose (pH 7.3 with KOH). Electrophysiological recordings were obtained using an Axopatch 200B amplifier (Axon Instruments), connected to a 1550B series Digidata (Molecular Devices). Electrodes had a resistance of 1–1.5 MΩ when filled with a solution containing (in mM): 130 NaCl, 5 KCl, 10 HEPES, 1 CaCl_2_, 1 MgCl_2_, and 10 TEA-Cl (pH 7.3 with NaOH). Mechanical stimulation was applied to membrane patches using high-speed pressure clamp (HSPC-1, ALA-scientific) through the pipette. The recordings were obtained at a holding potential of -80 mV with pressure steps from 0 to −60 mmHg (−5 mmHg increments).

For whole-cell recordings, HEK293T cells were kept in an extracellular solution containing (in mM) 10 glucose, 140 NaCl, 4 KCl, 2 MgCl_2_, 10 HEPES, 2 CaCl_2_, (pH 7.3). Electrodes were filled with a solution containing (in mM) 127 K-Gluconate, 10 NaCl, 5 KCl, 4 Mg-ATP, 0.4 Na-GTP, 5 Creatine P (Na), 1 CaCl_2_, 10 EGTA, 1 MgCl_2_, 10 HEPES, (pH 7.2). Mechanical stimulation was applied to cell membrane using a fire-polished glass probe. The movement of the probe was controlled by a piezoelectric microstage (PZ-150M, Burleigh). The probe had a velocity of 1 µm/ milliseconds and the stimulation was applied for 200 milliseconds. A series of mechanical steps of 0.5 µm increments were applied to the cells every 10 s. The recordings were obtained at a holding potential of −60 mV. For both cell-attached and whole-cell recordings, Yoda1 treatment consisted of the addition of Yoda1 to the electrode solutions at a final concentration of 30 µM. Recordings were acquired on Axon pCLAMP 10.6 (Molecular Devices) and analysed on Clampfit 10.6 (Molecular Devices). The protocol and fitting equation (when applicable) used are specified in each figure.

Recordings were acquired on Axon pCLAMP 10.6 (Molecular Devices) and analysed on Clampfit 10.6 (Molecular Devices). The protocol and fitting equation (when applicable) used are specified in each figure.

### 4.4. Single-Nuclei RNA Sequencing

Five hearts were dissected and placed in cold PBS with heparin (5 mg/mL). Atria and outflow tracts were removed and the ventricles were opened and washed in fresh cold PBS with heparin (5 mg/mL). All ventricles were pooled into a single Eppendorf tube, excess media was removed, and the samples were flash-frozen in liquid nitrogen and stored at −80 °C. Nuclei were isolated as described in the 10× Genomics protocol (CG000375.Rev C). Sorting was performed on a BD FACS Melody. Nuclei suspensions were loaded on a Chromium controller (10× Genomics, Pleasanton, CA, USA) to generate single-nuclei gel beads in emulsion (GEMs). Single-nuclei RNA-Seq libraries were prepared using Chromium Single cell 3′RNA Gel Bead and Library Kit v3.1. GEM-RT was performed in a C1000 Touch Thermal cycler with 96-Deep Well Reaction Module (Bio-Rad): 53 °C for 45 min, 85 °C for 5 min; held at 4 °C. After RT, GEMs were broken and the single-strand cDNA was cleaned up with DynaBeads MyOne Silane Beads (Thermo Fisher Scientific). cDNA was amplified using the C1000 Touch Thermal cycler with 96-DeepWell Reaction Module: 98 °C for 3 min; cycled 12: 98 °C for 15 s, 63 °C for 20 s, and 72 °C for 1 min; 72 °C for 1 min; held at 4 °C. Amplified cDNA products were cleaned up with SPRI select beads. Indexed sequencing libraries were constructed following these steps: (1) fragmentation, end-repair, A-tailing, and size selection with SPRIselect; (2) adapter ligation and cleanup with SPRIselect; (3) sample index PCR and size selection with SPRIselect. The barcoded sequencing libraries were quantified by quantitative PCR (KAPA Biosystems Library Quantification Kit for Illumina platforms). Sequencing libraries were loaded at 300 pM on an Illumina NovaSeq6000 using the following read lengths: 28 bp Read1, 8 bp I7 Index, 91 bp Read2 (experiment 1); and 28 bp Read1, 10 bp I7 Index, 10 bp I5 Index, 87 bp Read2 (experiment 2).

Image analyses and base calling were performed using the NovaSeq Control Software and the Real-Time Analysis component (Illumina). Demultiplexing was performed using the 10× Genomics software Cellranger mkfastq (v6.0.1), a wrapper of Illumina’s bcl2fastq (v2.20). The quality of the raw data was assessed using FastQC (v0.11.8) from the Babraham Institute and the Illumina software SAV (Sequencing Analysis Viewer). FastqScreen (v0.14.0) was used to estimate the potential level of contamination.

Alignment, gene expression quantification, and statistical analysis were performed using Cell Ranger count on Danio rerio’s transcriptome GRCz11 (sequences and annotation were downloaded from Ensembl! on 24 July 2019). In order to discard ambient RNA falsely identified as cells, Cell Ranger count was run a second time with the option --force-cells to force the number of cells to detect. For this experiment, 5 hearts were used resulting in 648 analysed nuclei with a median of 722 genes detected per nuclei.

### 4.5. Cardiac Performance Analysis

Tg(myl7:EGFP) embryos were exposed to 0.2 mM N-phenylthiourea (PTU) from the epiboly stage. At 4 dpf, larvae were treated with either DMSO (vehicle) or 50 µM Yoda1 (Selleckchem) in embryo medium for 30 min, anaesthetized using tricaine methanesulfonate at 168 mg/mL and mounted in low-melting point agarose. 1 min videos at 67 frames/second were recorded using a Zeiss Discovery v20 stereomicroscope and analysed using ZeCardio^TM^ software (ZeClinics).

### 4.6. Optical Mapping

Prior to voltage change recording, 4 dpf zebrafish larvae were incubated with DMSO or 50 µM Yoda1 (Selleckchem) and blebbistatin (8.5 µM, Selleckchem) (to avoid contractions) in embryo medium for 1 h. Larvae were then incubated in the dark for 40 min with the voltage-sensitive probe di-4-anepps (9 µM; ThermoFisher) with DMSO or Yoda1 and blebbistatin. Next, the larvae were immobilized in the lateral position in low-melting-point agarose, exposing their left side. Cardiac signals were then recorded for 30 to 60 s with an acquisition frequency of 100 frames/s using a 150 W halogen lighting system (SciMedia) as an excitation source, coupled with a 531/50 nm excitation filter, a 580 nm dichroic mirror, and a 580 long-pass emission filter. For the acquisition, the Microcam3 camera with a sensor of 17.6 × 17.6 mm (SciMedia; Costa Mesa, CA) was coupled to two optics: the SDF PLAN FLUOR 0.3 X stereo microscope objective mounted as an eyepiece lens (Olympus) and the Plan Apo 5.0 X/0.50 LWD mounted as magnification lens (Leica). Such a setting allows for an optic magnification of 16.6 X. The Microcam3 camera was connected to the BV Workbench software (ver 2.7.2, SciMedia; Costa Mesa, CA, USA) to allow recording and successive analysis of the fluorescent signals. Before acquisition, the temperature inside the petri dish was recorded and maintained at about 27 °C by a warming chamber (QE-1, Warner Instruments, Hamden, CT) including a temperature sensor, connected with a dual automatic temperature controller (TC-344B, Warner Instruments).

### 4.7. ECG Measurement

ECG measurements were performed in accordance with local approval (APAFIS#2021021117336492-32684) and as described in Arel and Rolland et al., [19] using the exposed heart configuration. Once a 30 s ECG recording was obtained, Yoda1 or DMSO were added to the water at a final concentration of 50 µM for one hour and ECG was recorded again for 30 s. The recordings were processed using LabChart Pro v8 software (AD Instruments) and its ECG analysis module.

### 4.8. Real-Time Quantitative PCR (RT-qPCR)

RNA was extracted from three different plates of HEK293T cells untransfected or transfected with pIRES2-GFP-*piezo1a* or pIRES2-GFP-*piezo1b* using Trizol/chloroform. cDNA was obtained after reverse transcription using a First strand cDNA synthesis RT-PCR kit (Roche), and quantitative PCR was performed using SYBR Green (Roche) and a LightCycler 480 system (Roche). The primer sequences used are as follows:

*GAPDH* Forward: 5′ TGCACCACCAACTGCTTAGC 3′

*GAPDH* Reverse: 5′ GGCATGGACTGTGGTCATGA 3′

*piezo1a* Forward: 5′ ATGACATAGGGCCCAGTGGA 3′

*piezo1a* Reverse: 5′ TGTCAGCCAGCTGTGATACG 3′

*piezo1b* Forward: 5′ CACCGGTGCTGATAGAGCAA 3′

*piezo1b* Reverse: 5′ CCTCACTGCTGTTAGCGGAA 3′

### 4.9. Data Analysis

Data were processed in GraphPad Prism 9.2.0 (GraphPad). Data are expressed as mean ± SEM, with the exception of the ECG graph, showing only paired individual measurements. Patch clamp data were analysed using either two-way ANOVA or *t*-test, as specified in the figure legends. T-test analysis was used to analyse cardiac performance (unpaired), optical mapping (unpaired), and ECG measurements (paired). RT-qPCR data for housekeeping gene expression were analysed using one-way ANOVA.

## 5. Conclusions

Here, we have shown that Yoda1-induced *PIEZO1* activation results in cardiac arrhythmias in zebrafish larvae. We speculate that Yoda1 treatment could be mimicking the effects of pathological situations such as atrial stretch and increased arterial blood pressure, which can activate *PIEZO1* and result in similar cardiac arrhythmias. Based on this, we believe it is possible that *PIEZO1* genetic variants with gain-of-function properties could be responsible for cardiac arrhythmias in humans. In summary, our data encourage *PIEZO1* genetic sequence analysis for patients suffering from hereditary cardiac arrhythmias.

## Figures and Tables

**Figure 1 ijms-24-06720-f001:**
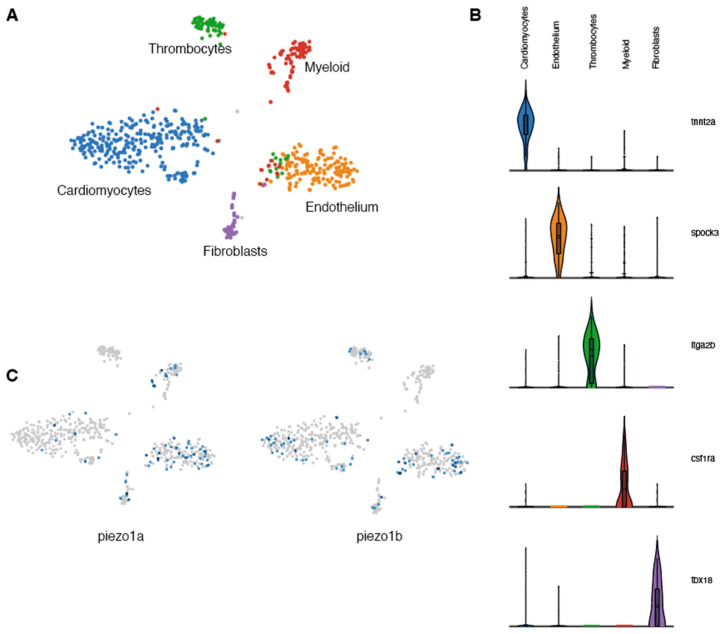
Single-nuclei RNAseq analysis of zebrafish Piezo expression in the heart. (**A**) UMAP clusters of the different populations of cells identified in zebrafish hearts. (**B**) Violin plots showing the 5 genes used to characterize the different cell populations. (**C**) UMAP plots indicating the cells expressing *piezo1a* or *piezo1b*.

**Figure 2 ijms-24-06720-f002:**
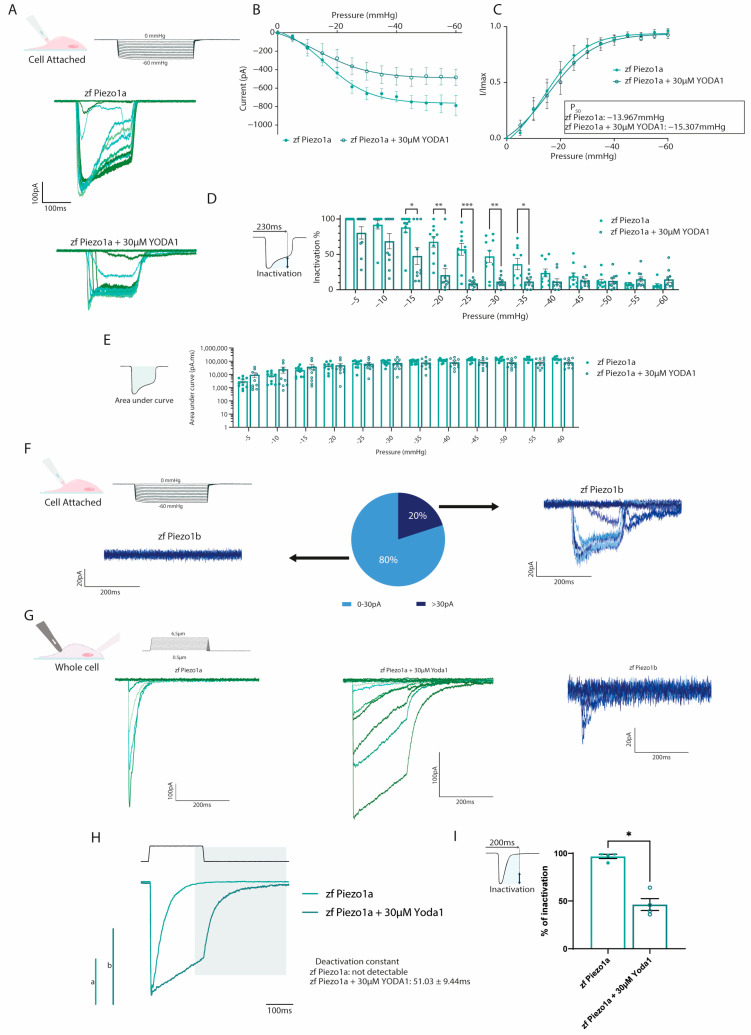
Electrophysiological recordings of zebrafish Piezos. (**A**–**D**) Cell-attached recordings obtained from HEK293T transfected with pIRES2-EGFP-*piezo1a*. n = 10 for each condition. (**A**) Representative zfPiezo1a traces obtained in untreated cells or cells treated with 30 µM Yoda1. (**B**) Current–pressure relationship. (**C**) I/Imax normalization fitted with Boltzmann equation. The average P_50_ was calculated from the fitting curve of each condition. (**D**) Proportion of inactivated channels 230 milliseconds after the beginning of a sweep (still under mechanical stimulation). (**E**) Area under curve calculated for cell-attached recordings. (**F**) Heterogeneity of the currents recorded in the cell-attached configuration in zf Piezo1b-transfected cells. (**G**) Representative whole-cell recordings obtained in cells transfected with either pIRES2-EGFP-*piezo1a* untreated (zf Piezo1a) or treated with 30 µM of Yoda1 (zf Piezo1a + 30 µM Yoda1) or pIRES2-EGFP-*piezo1b* (zf Piezo1b) and stimulated using mechanical indentations. (**H**) Representative deactivation current observed in the zf Piezo1a transfected cells treated with 30 µM of Yoda1. a,b: scale bars of zf Piezo1a (a) and zf Piezo1a + 30 µM Yoda1 (b), 1000 pA. (**I**) Proportion of current inactivated 200 milliseconds after the beginning of a sweep (still under mechanical stimulation). Data are presented as mean ± SEM. Statistical significance was assessed using two-way ANOVA followed by a Dunnett post hoc test for cell-attached recordings (**B**–**E**) and by *t*-test for the inactivation percentage of whole-cell recordings (**I**). *: *p*-value < 0.05, **: *p*-value < 0.01, ***: *p*-value < 0.001.

**Figure 3 ijms-24-06720-f003:**
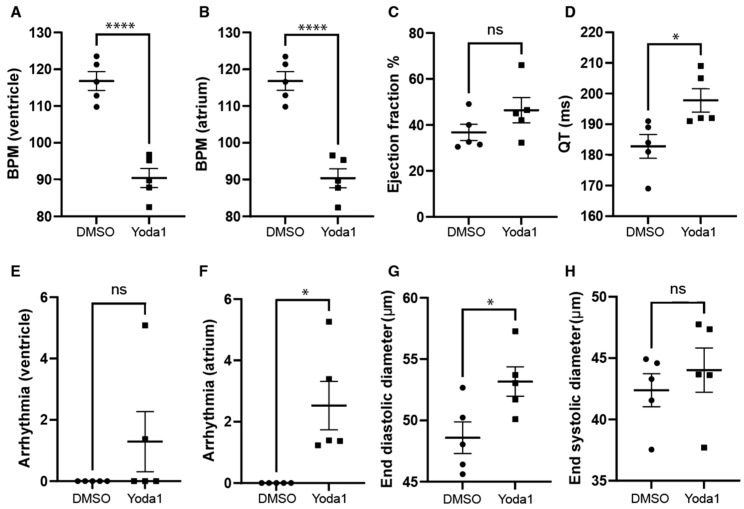
Cardiac physiology of Yoda1-treated larvae. (**A**) Ventricular rate. (**B**) Atrial rate. (**C**) Ejection fraction. (**D**) QT interval. (**E**) Ventricular arrhythmia. (**F**) Atrial arrhythmia (**G**) End diastolic diameter. (**H**) Ventricle maximum fractional length. n = 5. Statistical significance was assessed by independent *t*-test and Bonferroni correction. *: *p*-value < 0.05, ****: *p*-value < 0.0001.

**Figure 4 ijms-24-06720-f004:**
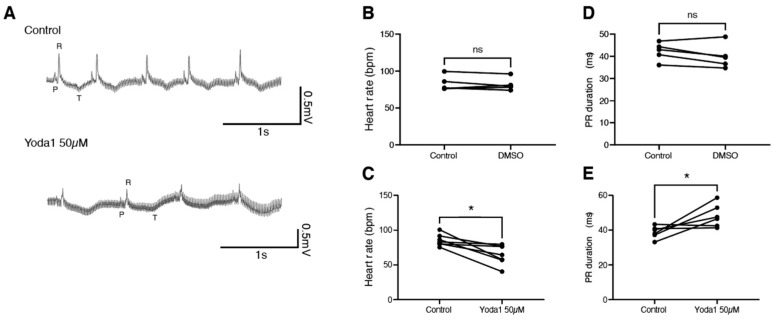
ECG recordings of Yoda1-treated adult zebrafish. (**A**) Representative recordings obtained from the same fish before and after Yoda1 treatment. (**B**) Heart rate (in beats per minute, bpm) measured before and after a 1 h DMSO treatment; n = 5 (**C**) Heart rate measured before and after a 1 h Yoda1 treatment (50 µM); n = 7. (**D**) PR duration (in ms) measured before and after a 1 h DMSO treatment; n = 5 (**E**) PR duration measured before and after a 1 h Yoda1 treatment (50 µM); n = 6. Statistical significance was assessed by paired *t*-test. *: *p*-value < 0.05.

**Figure 5 ijms-24-06720-f005:**
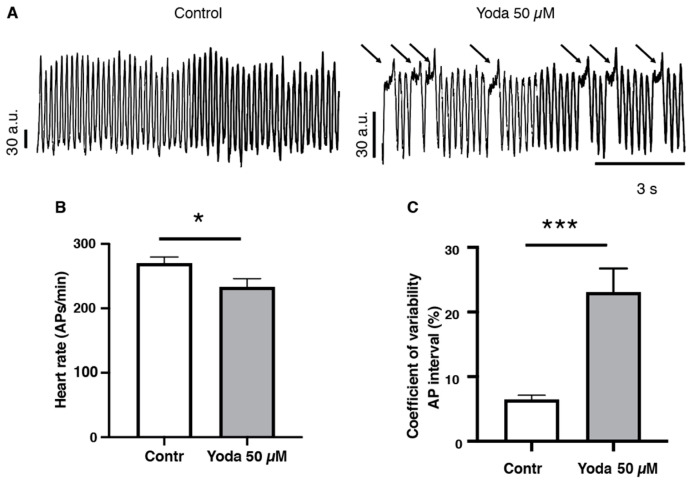
Optical mapping recordings of Yoda1-treated larvae. (**A**) Traces of optical action potentials recorded in the ventricular region of the larvae’s hearts, in control (DMSO) or after prolonged exposure to Yoda (50 µM). Arrows indicated prolonged ventricular depolarization. (**B)** Heart rate (ventricular action potentials per minute) and (**C**) coefficient of variability of the action potential interval in control (DMSO) and after prolonged exposure to Yoda 50 µM (n = 19 and 23, respectively). *: *p* < 0.05 and ***: *p* < 0.001 by unpaired *t*-test.

## Data Availability

Data will be made available upon request by the corresponding authors.

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
