# Peer review of "Prolonged Piezo1 Activation Induces Cardiac Arrhythmia"

_ijms, 2023, doi:10.3390/ijms24076720_

Round 1

Reviewer 1 Report

Comments for the Authors

The manuscript from Laura Rolland, Chris Jopling, and Adèle Faucherre entitled ‘’ Prolonged Piezo1 activation induces cardiac arrhythmia.’’ investigates the consequence of hyperactivation of the mechanical-activated Piezo1 channel in the genesis of cardiac arrhythmias.

It is an interesting manuscript; however, I have a few comments concerning this study.

Major comments:

1-     Figure 3D. Knowing that the heart rate is altered after treatment, it will be interesting also to calculate and present the corrected QT.

2-     In the section ‘’Data analysis’’, add which statistical test has been used.

3-     Why the ‘’conclusions’’ part is after the ‘’Materials and Methods’’ part and not after the discussion?

4-     From my point of view, the discussion needs to be rewritten and looks like a repetition of the abstract. In contrast, the conclusion looks like the discussion. Please rewrite the discussion and the conclusion. In the discussion, you should propose a potential mechanism explaining your observation and summarize the findings. Why not discuss the observations done previously on Piezo1, hypertrophy, and the potential role of the TRPM4 channel?

5-     If Yoda1 decreases the peak current (Fig.2b) but also decreases the inactivation kinetics of Piezo1 current (Fig.2D), it will be interesting to calculate the area under the curve to conclude concerning the effect mediated by Yoda1 on the Piezo1 channel.

6-     Why the kinetic of inactivation is different between the two approaches (cell-attached versus whole-cell)?

7-     If in whole-cell configuration condition, the inactivation kinetics of Piezo1 channel is faster than in cell-attached and knowing that Yoda1 treatment leads to the decrease of this parameter, what is the effect of Yoda1 on zfPiezo1a current expressed in heterologous expression system using the whole-cell configuration?

8-     What is the level of expression of Piezo1 and 2 in a heterologous expression system?

9-     Please explain how the drug Yoda1 was applied in the different sets of experiments and for how long?

Minor comments:

1-     The part between lanes 74 and 81 should be removed.

2-     The term ‘’subsection’’ lane 85 should be removed.

3-     Lane 117. The term ‘’ zfPiezo1a’’ is not introduced before. Why ‘’zf’’?

4-     Be consistent. Always add a space between a figure and the unit (e.g., 50 µM instead of 50µM).

5-     It should be CaCl2 and MgCl2 instead of CaCl2 and MgCl2.

6-     Sometimes the pH is written pH = 7.2 and after pH = 7.30. Be consistent.

7-     Lane 303. Please add the concentration of heparin.

8-     Lane 273, Please add the information for the Penicillin/Streptomycin activities.

9-     Lane 305. You wash the sample with what? Please add this information.

10- Lane 306. It should be -80 ºC instead of -80C.

11- Lane 363. It should be 27 ºC instead of 27 C

12- Be consistent with the time units. Min., sec., and h. or minutes, seconds, and hours.

13- Lanes 408-413. This part should be removed.

Reviewer 2 Report

By this manuscript, authors used zebrafish model to investigate cardiac electrophisiology of Prolonged Piezo1 activation in the presence7absence of Yoda1.

Altough I have no concern on data and results which are clear and well presented; in my opinion this manuscipt should be revise to improve its clearness to the reader.

-Yoda1 (role, interactions, etc.) should be mentioned also in introduction section to let readers better undertand its role

- Revise paragraph 1 removing manuscript drafting instructions; the same in Data Availability Statement

- Revise manuscript so that Piezo1 is written the same way

- Revise reference number and corretc positioning of [9] and [10]

1.  By this manuscript, authors used zebrafish model to investigate cardiac electrophisiology of prolonged Piezo1 activation in the presence/absence of Yoda1.   2.  Even tough interesting, topic is not so original; this paper does not fill a specific gap in the field of molecular cardiology   3.  Effect of Yoda1 on the hyper activation of Piezo1   4.  Considering experimental methodology I have no concerns. Manuscript’s methods description is lacking of some fundamental details like description of Yoda1 treatment in different experiments and more information about use of heparin   5. Conclusions are consistent with data presented and  they address to the question posed   6. References are ok; revise number sequence to be more consistent (e.g. references 9 and 10 are not reported after 8 in the text)   7. As described manuscript should be improved to increase readability. Section to be modified are    - introduction: remove instruction o draft the paper; introduce Yoda1; be consistent while writing Piezo1   - methods: include what is stated in point No.4   - manuscript instruction are present also in section Data Availability Statement   - general revision is strongly suggested

Round 2

Reviewer 1 Report

Thanks to the authors for the review. I have only two final comments.

1- Lane 274 it should be ''resulting'' instead of ''reulsting''.

2-do a final check for the spaces between numbers and units, particularly in legends.

Author Response

1- Lane 274 it should be ''resulting'' instead of ''reulsting''.

Thank you for identifying this typo which we have rectified

2-do a final check for the spaces between numbers and units, particularly in legends.

We have checked the spaces and changed them where necessary